# Effect of Homozygous Sickle Cell Anemia on Perinatal Outcomes: A Retrospective Cohort Study

**DOI:** 10.3390/jcm14061967

**Published:** 2025-03-14

**Authors:** Ahmet Zeki Nessar, Fikriye Işıl Adıgüzel, Sefanur Gamze Karaca, Yusuf Dal, Zeynep Küçükolcay Coşkun, Ayhan Coşkun

**Affiliations:** 1Division of Perinatology, Department of Obstetrics and Gynecology, Mersin University Faculty of Medicine, Mersin 33110, Turkey; drkaracasefanur@gmail.com (S.G.K.); dryusufdal@gmail.com (Y.D.); drayhancoskun@hotmail.com (A.C.); 2Department of Obstetrics and Gynecology, University of Health Sciences, Adana City Hospital, Adana 01370, Turkey; aze_isil@hotmail.com; 3Department of Obstetrics and Gynecology, Mersin University Faculty of Medicine, Mersin 33110, Turkey; drzkucukolcay@gmail.com

**Keywords:** perinatal outcomes, pregnancy, sickle cell anemia

## Abstract

**Backgrounds**: Sickle cell anemia (SCA) is a multisystemic disorder that causes hemolytic anemia and impaired tissue perfusion due to sickling of red blood cells. Although there is a belief that adverse perinatal outcomes are frequent in pregnant women with SCA, this association has not been clearly established. The aim of this study was to compare the perinatal outcomes of women with homozygous mutated SCA who gave birth with those without the mutation. **Methods**: The study included 26 SCA patients with homozygous mutation and 108 pregnant women without mutation who gave birth in our center. Demographic and obstetric data, laboratory findings, and fetal findings of both groups were compared. **Results:** Statistically significant differences were found between the groups in terms of maternal age, body mass index (BMI), gravida, and parity (*p* ≤ 0.001, *p* = 0.035, *p* ≤ 0.001, *p* ≤ 0.001, respectively). Mean corpuscular volume (MCV), mean corpuscular hemoglobin (MCH), red blood cell count (RBC), hemoglobin (Hb), and hematocrit (Hct) values were significantly lower in the SCA group. We also observed that more blood transfusions were performed during pregnancy and the postpartum period in the SCA group. Low birth weight, more neonatal intensive care unit admissions, and a higher cesarean section rate were present in the SCA group. During pregnancy, women with SCA were most frequently admitted to the hospital for acute painful crises. Preeclampsia was not more common in the SCA group. **Conclusions:** SCA carries serious risks for the mother and fetus during pregnancy. Therefore, the relationship between the disease and pregnancy requires more detailed research.

## 1. Introduction

SCA is a multisystemic disease that occurs when glutamic acid is replaced by valine at position 6 in the hemoglobin beta chain, causing sickling in red blood cells. This mutation results in the production of an aberrant hemoglobin known as hemoglobin S. Red blood cells with hemoglobin S exhibit reduced flexibility and increased stickiness. Deformed red blood cells (sickle cells) encounter challenges traversing capillaries, resulting in interrupted blood flow [1]. The disease presents a complex clinical picture with chronic hemolytic anemia and vaso-occlusive attacks. Sickle cell anemia was first described by JB Herrick in 1910 as a case report in a young patient from Grenada [2]. The disease is common in Africa, the Middle East, the Mediterranean region, and India [3]. Turkey is one of the geographical regions where SCA is common. Although prenatal detection of homozygous genotypes is possible through premarital screening programs, preimplantation genetic diagnosis (PGD), which informs carrier parents of the genotype of the embryo, or genetic diagnostic tests such as chorionic villus sampling (CVS) and amniocentesis, which can be performed during pregnancy, a large number of babies continue to be born with homozygous genotypes, mostly in poor countries with limited access to health services. On average, 300,000 babies are born with sickle cell anemia every year, and this number is expected to increase [4]. Despite the condition being recognized nearly a century ago, the advancement of treatment modalities has progressed at a sluggish pace. Allogeneic hematopoietic stem cell transplantation and genome-based therapeutic strategies have emerged as promising modalities for patient treatment in recent years [5]. While hydroxyurea remains the primary medication for the condition, the emergence of novel therapeutics such as L-glutamine, crizanlizumab, and voxelotor indicates the potential for expanded treatment avenues [6].

Pregnant women with sickle cell disease face more adverse perinatal outcomes [7]. Pregnant women with SCA have an increased risk of abortion, preterm labor, IUGR, and stillbirth [8]. A meta-analysis has shown that the risk of maternal mortality is 6 times higher and the risk of preeclampsia is 2.5 times higher in pregnant women with SCA. It has also been found that these women have an increased risk of premature birth, stillbirth, and fetal growth retardation [9]. The most common cause of morbidity during pregnancy is painful crises. These crises occur due to vaso-occlusion in the organs due to increased sickling in the presence of precipitating conditions, such as stress, heavy physical activity, dehydration, and cold weather conditions, and are also known as vaso-occlusive crises (VOC). Although the use of hydroxyurea in SCA patients has been shown to reduce the frequency of painful crises, its use in pregnancy is out of the question because the drug is teratogenic [10,11]. However, there are still clinical studies that show that the use of hydroxyurea in pregnant women with severe symptoms in the second or third trimester outweighs the benefits it provides to the patient [12]. Prophylactic transfusion of SCA patients during pregnancy remains controversial. Although this approach was initially thought to be beneficial, a randomized controlled trial found no effect on perinatal outcomes [13]. ACOG does not recommend prophylactic blood transfusion except for medical indications because of the risks involved [11]. All these show that pregnant women with SCA should be followed up in multidisciplinary centers. In our study, we compare the perinatal outcomes of pregnant women with homozygous Hb SS mutation who gave birth with the outcomes of healthy pregnant women.

## 2. Materials and Method

### 2.1. Patients and Data Collection

In Mersin, which is located in the south of Turkey and is among the regions where sickle cell anemia is most common, 7864 live births occurred in the Mersin University Faculty of Medicine Hospital, a tertiary health center, between January 2012 and March 2024. During the same period, 11,220 women applied to various clinics of the hospital with the diagnosis of sickle cell syndromes, such as HbSS, HbSC, and HbS-beta thalassemia.

In total, 320 women were examined in the gynecology and obstetrics clinic due to pregnancy. We found 45 pregnant women diagnosed with homozygous HbSS, 14 of whom had pregnancies that resulted in abortion or stillbirth; 5 of the remaining 31 patients had incomplete records and had discontinued follow-ups at our center, and their births probably occurred in other centers. Patient files and the hospital’s digital database were examined, and 26 women with homozygous sickle cell disease who had complete records and met the inclusion and exclusion criteria were selected as the study group. In total, 108 healthy pregnant women without hemoglobinopathy or other hematological diseases were selected as the control group by computer randomization method. (Figure 1). The patients in the study group actually consist of patients who are followed up in the hematology clinic in our center. The patient files were examined, and it was determined that the definitive diagnoses were made by high-performance liquid chromatography (HPLC). Due to the prevalence of hemoglobinopathy in Turkey, it has been made mandatory for couples to have hemoglobin electrophoresis before marriage. Within the scope of the program, those with abnormal findings in hemoglobin electrophoresis are consulted by a hematologist. The pregnant women in the control group were selected from women whose tests, such as this screening test and complete blood count, were normal. All patients delivered in our hospital between January 2012 and March 2024 at 23 weeks of pregnancy or more. Pregnant women were longitudinally followed until delivery by our multidisciplinary team, involving gynecologists and obstetricians from the Perinatology Clinic of the Mersin University Hospital. During pregnancy, SCA patients were clinically evaluated every two weeks by a multidisciplinary team. Complete blood count, biochemical tests, standard urine analysis, and proteinuria were determined at each visit site. During prenatal visits, women were checked for gestational diabetes, pregnancy-induced hypertension, Hb levels, signs of urinary tract infection, and proteinuria.

### 2.2. Obstetric Follow-Up, Timing of Delivery and Demographic, Perinatal and Fetal Outcomes

The demographic data like maternal age, body mass index (BMI), gravida, parity, abortion, stillbirth history, and ectopic pregnancy history were recorded, and in addition, red blood cell (RBC) count, hemoglobin concentration (Hb), hematocrit (Hct), white blood cell count, platelet count, mean corpuscular volume (MCV), mean corpuscular hemoglobin (MCH), mean corpuscular hemoglobin concentration (MCHC), and leukocyte values at the time of admission to the hospital for delivery were also analyzed. Complete blood count parameters were determined using ADVIA 2120i (Siemens Healthcare, Diagnostics Inc., Tarrytown, NY, USA), and biochemical parameters were determined using Beckman Coulter AU680 and AU480 (Beckman Coulter, Inc., Brea, CA, USA) devices.

As prenatal outcomes, the need for transfusion during pregnancy or postpartum, complications such as preterm labor, intrauterine growth retardation (IUGR), preeclampsia, VOC, gestational age at delivery, delivery type (vaginal delivery or cesarean section) of the patients, and controls were recorded. IUGR is broadly defined as an estimated fetal weight < 10th percentile for gestational age with pathological Doppler findings. Those without pathological Doppler findings (small-for-gestational age) were not evaluated as a complication [14]. Prematurity is defined as a birth that occurs before 37 completed weeks of gestation [15]. The American College of Obstetricians and Gynecologists criteria published in 2020 were used to determine the diagnosis of preeclampsia. For the diagnosis of preeclampsia, systolic blood pressure ≥140 mmHg and/or diastolic blood pressure ≥90 mmHg in two measurements 4 h apart after the twentieth week of pregnancy in a woman without previous hypertension and accompanied by various multisystem disorders with or without new-onset proteinuria (platelet count < 100,000 × 10^9^/L, liver enzymes elevated to twice the upper limit of normal concentration, unexplained severe persistent right upper quadrant or epigastric pain, renal failure (serum creatinine ≥ 1.1 milligrams per deciliter (mg/dL), or doubling of serum creatinine concentration in the absence of renal disease), pulmonary edema, or new-onset headache unresponsive to acetaminophen and unexplained) were used [16]. VOC was characterized by bone pain (i.e., in the extremities, hips, or back) or abdominal pain without another clinical explanation [17]. Given the documented prothrombotic state in sickle cell patients and the established independent risk of venous thromboembolic events associated with sickle cell disease during pregnancy, enoxaparin was administered at a prophylactic dosage of 4000 U daily at the initial visit following a positive pregnancy test [18].

Birth weight, birth weight percentile, APGAR score at 1 min and 5 min, umbilical artery pH, and neonatal intensive care unit (NIUC) admission requirements were recorded as fetal outcomes. Term or preterm babies diagnosed with IUGR were examined by a pediatric specialist immediately after birth and transferred to the NICU if a decision were made for NICU admission.

### 2.3. Statistical Analysis

We used the Shapiro–Wilk test to find out whether continuous data were normally distributed. While mean ± standard deviation was used for normally distributed continuous variables, median [25–75%] was used for others. Categorical variables were collected as numbers and percentages. Parametric comparisons were made using the independent sample *t*-test, and nonparametric comparisons were made using the Mann–Whitney U test. Statistics were considered significant when *p* < 0.05.

## 3. Results

The study included 26 patients diagnosed with SCA and 108 patients as a control group. Table 1 displays the demographic features and hematological values of the patients. While the groups showed statistically significant differences in terms of maternal age, body mass index (BMI), gravida, and parity, there was no statistical difference in terms of miscarriage, stillbirth history, and ectopic pregnancy history.

Women with SCA had significantly lower Hb, Hct, and red blood cell counts and higher MCV, MCH, and platelet counts compared to the control group. In the statistical evaluation made between the groups, no significant difference was detected in MCHC and leukocyte values; however, statistical differences were detected between the groups with respect to Hb, Hct, MCV, RBC, and platelet values.

The perinatal and fetal outcomes are shown in Table 2. During pregnancy, 20 (76.9%) patients with SCA and 1 (0.9%) patient from the control group required transfusion, and in the postpartum period, 15 (57.7%) patients with SCA and 1 (0.9%) control patient required transfusion. When the complication rates were examined, 3 (11.5%) preterm labor, 5 (19.2%) IUGR, and 6 (23.1%) VOC were observed in the SCA group, while 8 (7.4%) preterm labor, 6 (5.6%) IUGR, and 1 (0.9%) were observed in the control group. There is a statistically significant relationship between the groups in terms of complications (*p* < 0.001). In order to determine which complication caused this difference, a pairwise ratio comparison was made. There is a significant difference between the groups in terms of no complications (*p* < 0.001). There is a significant difference between the groups in terms of VOCs (*p* < 0.001). There is a statistically significant difference between the groups in terms of IGUR (*p* = 0.024). While there were statistically significant differences in terms of fetal outcomes such as birth weight, birth weight percentile, and NICU admission requirement, there was no difference between the groups in terms of Apgar scores and umbilical pH values. However, when covariance analysis was performed for numerical perinatal outcomes by taking parity and maternal age as covariates, the difference between SCA–control groups continued for all numerical variables except Apgar at 5 min. When the effect of maternal age and parity was adjusted for Apgar at 5 min, the *p*-value was 0.149. Log-linear models were used for categorical variables, and when the effect of maternal age and parity was adjusted, the complication variable lost its significance.

## 4. Discussion

SCA, which is caused by abnormal hemoglobin β chain production as a result of a point mutation, is a multisystem disease with many unique complications. The disease is common in Africa, the Middle East, the Mediterranean region, and India [19]. In individuals with homozygous (Hb SS) mutation, sickling caused by abnormal hemoglobin triggers VOC events. The resulting cytokines and inflammatory mediators cause tissue hypoxia and organ damage [20]. In addition, abnormally shaped erythrocytes undergo phagocytosis and form the basis of hemolytic anemia. As a result, acute and chronic complications such as hemolytic anemia, acute chest syndrome, susceptibility to infections, acute pain crisis, stroke, acute splenic secretion crisis, aplastic crisis, hepatic and biliary tract complications, avascular necrosis, pulmonary hypertension, growth, and developmental retardation negatively affect the quality of life and expectation of patients. Currently, in addition to medical methods in the treatment of the disease, there are options such as blood transfusion and bone marrow transplantation [21].

The effect of sickle cell anemia on reproduction and fertility is not clearly known. It has been reported that these patients have a higher age at first pregnancy and fewer children due to the late onset of menarche [22]. In the study by M’Pemba-Loufouma et al., a delay in puberty and sexual maturation was also mentioned in girls with homozygous SCA [23]. Carvalho et al. observed delayed age at menarche in those with HbSS genotype when compared with other sickle cell genotypes [24]. In our study, the high average age and low parity in the SCA group seem to be consistent with this information.

VOCs resulting from sickling in red blood cells in pregnant women with sickle cell anemia may affect all organs and systems as well as trigger infarction, villous necrosis, and fibrosis in the placenta, thus reducing uteroplacental circulation and leading to chronic fetal hypoxia, preterm birth, fetal growth retardation, and adverse fetal outcomes [25]. Studies have shown that pregnant women with homozygous sickle cell disease (SCA) experience more adverse perinatal outcomes. In a study by Lewis et al., it was determined that homozygous SCA patients had higher rates of spontaneous abortion, low birth weight, preeclampsia, urinary tract infection, maternal death related to childbirth, and lower live birth rates [26]. In addition, in the study conducted by Serjeant et al., it was observed that complications such as preterm birth and placental retention were more common in SCA pregnant women compared to healthy controls [27]. In our study, we observed that preterm birth, intrauterine growth retardation (IUGR), and low birth weight were higher in the SCA group.

In our study, we observed that patients in the SCA group received more blood transfusions than the control group, both during pregnancy and in the postpartum period. Blood transfusion is a lifesaving treatment, especially in those with severe anemia. Although there are studies supporting the positive effect of prophylactic blood product transfusion on maternal and fetal outcomes in pregnant SCA patients [28], a randomized controlled study by Koshy et al. found that it had no effect on perinatal outcomes [13].

There are studies indicating an increased risk of preeclampsia in SCA patients. Howard et al., in their article, suggest that pregnant women with SCA should be considered high-risk pregnancies in terms of preeclampsia and eclampsia based on retrospective studies [29]. In our study, we did not observe a difference between the groups in terms of preeclampsia development. Our findings coincide with the study by Taylor et al. [30]. The relationship between SCA and preeclampsia–eclampsia remains unclear, and we believe that studies are needed in this area.

We believe that the higher anemia observed in the SCA group in hematological parameters in our study is due to the possible connection between SCA and anemia in pregnancy [31].

The development of complications such as IUGR, prematurity, stillbirth, and fetal distress in pregnant women with SCA has been mentioned in many studies. Ganesh et al. also mentioned this relationship in their study evaluating data from India [8]. Similarly, Aghamolaei et al. found in a meta-analysis that SCA poses a risk for fetal complications such as IUGR, stillbirth, and prematurity [32]. In our study, we also found that birth weight was lower and the rate of cesarean section was higher in the SCA group. Diagnosis of intrauterine growth retardation has an effect on the type of delivery and causes an increase in the cesarean section rate [33]. In this case, it is clear that low birth weight increases the need for a neonatal intensive care unit.

In our study, we found that pregnant women in the SCA group were most commonly admitted to hospital due to acute pain crises. Pain crisis is the most common cause of maternal morbidity during pregnancy. Painful crises are also the main cause of hospitalization. VOCs were found to be the most common cause of hospitalization in pregnant women with SCA in the study by Silva et al. [34]. Treatment includes oral, intramuscular, or intravenous analgesia, depending on the severity of pain, as well as oxygen administration to patients with low oxygen saturation [35].

The main limitation of our study is that it is a retrospective single-center study and the number of patients is small. The article does not thoroughly consider other potential influencing variables, such as socioeconomic status and maternal health behaviors, which could have a potential impact on our study results. We were also unable to compare the mean age at first pregnancy between the two groups, but we think that if these data were available, they would provide additional insight into the timing of reproduction in women with SCA and how this may affect perinatal outcomes. On the other hand, it is one of the rare studies conducted in Turkey in relation to SCA and pregnancy.

## 5. Conclusions

SCA is a multisystemic chronic disease that involves serious fetal and maternal complications. Despite widespread screening programs and preconceptionally diagnosis opportunities, thousands of individuals with SCA are born each year. Although there are studies on SCA and pregnancy, we believe that more comprehensive studies are still needed.

## Figures and Tables

**Figure 1 jcm-14-01967-f001:**
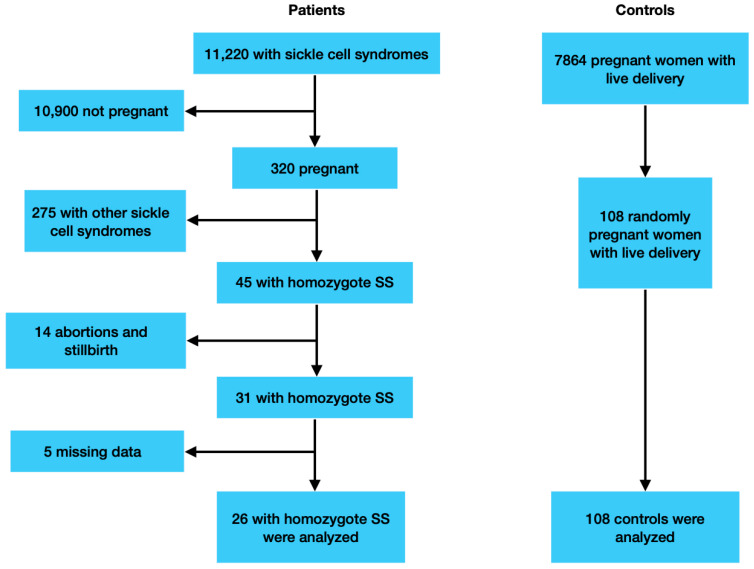
Patients’ selection flow chart.

**Table 1 jcm-14-01967-t001:** Demographic and laboratory measurements of the study participants.

	Patients (n = 26)	Control (n = 108)	*p*
Age (year)	36.50 ± 6.35	25.24 ± 6.48	<0.001
Gravida [min–max]	1 [1–6]	3 [1–9]	<0.001
Parity [min–max]	0 [0–3]	2 [0–5]	<0.001
Abortion [min–max]	0 [0–3]	0 [0–4]	0.496
Stillbirth history [min–max]	2 [1–2]	2 [1–2]	0.539
Ectopic pregnancy history [min–max]	0 [0–1]	0 [0–1]	0.781
Hb (g/dL)	8.71 ± 1.53	10.34 ± 1.54	<0.001
Hct (%)	25.56 ± 4.37	30.99 ± 3.85	<0.001
Hb F level (g/dL)	10.06 ± 7.09	-	
MCV (fL)	89.37 ± 7.32	82 ± 6.81	<0.001
MCH (pg)	30.69 ± 3.20	27.66 ± 3.46	<0.001
MCHC (g/dL) [min–max]	34 [28–39]	34 [29–37]	0.323
RBC (106/μL)	3 ± 0.63	3.7 ± 0.36	<0.001
Platelet (10^3^/μL)	388.34 ± 126.09	209.64 ± 60.85	<0.001
Leukocyte (10^3^/μL)	15.75 ± 5.39	15.22 ± 4.22	0.643
Neutrophil (%)	53.9 ± 3.1	55.8 ± 2.6	0.984

n, number of patients in the group; mg/dL: Milligrams per decilitre; fL, femtoliter; pg, picogram; μL, microliter.

**Table 2 jcm-14-01967-t002:** Perinatal and fetal outcomes of the study participants.

	Patients(n = 26)	Control(n = 108)	*p*
Transfusion during pregnancy			<0.001
(+)	20 (76.9%)	1 (0.9%)	
(−)	6 (23.1%)	107 (99.1%)	
Postpartum transfusion			<0.001
(+)	15 (57.7%)	1 (0.9%)	
(−)	11 (42.3%)	107 (99.1%)	
Complication			<0.001
(−)	12 (46.2%) ^a^	93 (86.1%) ^b^	
Preterm labor	3 (11.5%) ^a^	8 (7.4%) ^a^	
IUGR	5 (19.2) ^a^	6 (5.6%) ^b^	
VOC	6 (23.1) ^a^	0 (0%) ^b^	
Preeclampsia	0 (0%) ^a^	1 (0.9) ^a^	
Gestational age (week) at delivery	37.5 [35.75–38]	38 [37–39.75]	0.001
Delivery type			<0.001
Vaginal delivery	2 (7.7%)	34 (74.8%)	
Caesarean section	24 (92.3%)	74 (68.5%)	
Birth weight (g)	2552.19 ± 748.83	3222.82 ± 533.00	<0.001
Birth weight percentile (%)	29.62 ± 30.57	62.45 ± 27.34	<0.001
APGAR score at 1 min	9 [0–9]	9 [2–9]	0.030
APGAR score at 5 min	9 [0–10]	10 [6–10]	0.044
Umbilical artery pH	7.31 [7.10–7.43]	7.33 [7.15–7.46]	0.018
NICU admission			<0.001
(+)	9 (34.6%)	0 (0%)	
(−)	17 (65.4%)	108 (100%)	

VOC, vaso-oclusive crisis; NICU, neonatal intensive care unit; IUGR, intrauterine growth retardation. Different superscript letter denotes a significant difference between groups.

## Data Availability

Derived data supporting the findings of this study are available from the corresponding author.

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
