# Peer review of "Effect of Homozygous Sickle Cell Anemia on Perinatal Outcomes: A Retrospective Cohort Study"

_jcm, 2025, doi:10.3390/jcm14061967_

Round 1

Reviewer 1 Report

Comments and Suggestions for Authors

I would like to congratulate the authors on their fascinating work regarding this interesting report entitled “Effect of homozygous sickle cell anemia on perinatal outcomes: A retrospective cohort study”. This study compared the perinatal outcomes of women with homozygous mutated Sickle cell anemia (SCA) who gave birth with those without the mutation. The manuscript is well-written and the incorporated figure and tables make the study easy to follow.

1)      Abstract

-“Statistically significant differences were found between the groups in terms of maternal age, body mass index (BMI), gravida and parity.” I would suggest adding the exact p-value for each difference.

2)      Introduction

-“In our study, we planned this study to compare the perinatal outcomes of pregnant women with homozygous Hb SS mutation who gave birth with the outcomes of healthy pregnant women.

This sentence does not seem well. Please re-write is as “In our study, we compare the perinatal outcomes of pregnant women with homozygous Hb SS mutation who gave birth with the outcomes of healthy pregnant women.

-          References that are used in introduction are very old

-          Ref 1 (2001)

-          Ref 3 (2013)

-          Ref 5 (1988)

-          Ref 6 (1997)

I would suggest including references from articles published in the last 10 years.

3)      Material and methods

-How did you find the minimum number of patients that should be included in your study and which test did you perform to find it?

- In table one please also include the %NEUT type of leucocytes

4) Discussion

In discussion section you write in the text about ÇavuÅŸ et al. reference 15 ,, but this Reference belongs to:

Serjeant, G.R.; Loy, L.L.; Crowther, M.; Hambleton, I.R.; Thame, M. Outcome of pregnancy in homozygous sickle cell disease. Obstetrics and gynecology 2004, 103, 1278-1285, doi:10.1097/01.Aog.0000127433.23611.54.

Please make sure that all references are written well.

“Treatment includes oral, intramuscular or intravenous analgesia, depending on the severity of pain, as well as oxygen administration to patients with low oxygen saturation.”

A reference is missing to this text

Author Response

Dear reviewer,

Thank you very much for your suggestions. We believe that our article will be much more scientific after the changes we made. We have addressed all the comments as shown in the revised manuscript in yellow highlighted.

1)      Abstract

  • “Statistically significant differences were found between the groups in terms of maternal age, body mass index (BMI), gravida and parity.” I would suggest adding the exact p-value for each difference.

Answer: p-values were added.

2)     Introduction

  • “In our study, we planned this study to compare the perinatal outcomes of pregnant women with homozygous Hb SS mutation who gave birth with the outcomes of healthy pregnant women.” This sentence does not seem well. Please re-write is as “In our study, we compare the perinatal outcomes of pregnant women with homozygous Hb SS mutation who gave birth with the outcomes of healthy pregnant women.

Answer: This sentence was rewritten as ‘’In our study, we compare the perinatal outcomes of pregnant women with homozygous Hb SS mutation who gave birth with the outcomes of healthy pregnant women.’’

  • References that are used in introduction are very old

-          Ref 1 (2001)

-          Ref 3 (2013)

-          Ref 5 (1988)

-          Ref 6 (1997)

I would suggest including references from articles published in the last 10 years.

Answer: References that are used in introduction were changed.

3)      Material and methods

  • How did you find the minimum number of patients that should be included in your study and which test did you perform to find it?

Answer: Dear reviewer, this number is a single center data and includes all pregnant women with homozygous mutation among patients who were followed up with the diagnosis of sickle cell syndromes in the last 10 years.

  • In table one please also include the %NEUT type of leucocytes

Answer: %NEUT type of leucocytes were added to table 1.

4) Discussion

  • In discussion section you write in the text about ÇavuÅŸ et al. reference 15 ,, but this Reference belongs to:

Serjeant, G.R.; Loy, L.L.; Crowther, M.; Hambleton, I.R.; Thame, M. Outcome of pregnancy in homozygous sickle cell disease. Obstetrics and gynecology 2004, 103, 1278-1285, doi:10.1097/01.Aog.0000127433.23611.54.

Please make sure that all references are written well.

Answer: The reference was corrected.

  • “Treatment includes oral, intramuscular or intravenous analgesia, depending on the severity of pain, as well as oxygen administration to patients with low oxygen saturation.” A reference is missing to this text

Answer: The reference was added.

Reviewer 2 Report

Comments and Suggestions for Authors

Review of the Article: "Effect of Homozygous Sickle Cell Anemia on Perinatal Outcomes: A Retrospective Cohort Study"

This study provides valuable insights into the perinatal outcomes of women homozygous for sickle cell disease (SCA), a population known to be at higher risk for adverse pregnancy outcomes. The authors present important findings, comparing women with SCA to healthy controls and assessing various maternal and fetal outcomes. However, several aspects of the methodology, results presentation, and discussion could be strengthened to improve clarity and provide a more comprehensive understanding of the findings. Although the sample size is small, this study adds valuable insights to the existing literature on this topic.

Suggestions for Improvement:

1. Methodology:

The timing of blood tests is not clearly specified. It would be helpful to know at what gestational week these tests were performed and whether this timing was consistent across both the SCA and control groups.

The criteria for preterm labor and intrauterine growth restriction (IUGR) are not defined.

2. Results Section:

The presentation of results could be more concise. There is no need to present all the detailed statistical values from the tables in the text. A more streamlined narrative could improve readability. For example, the authors could summarize the main findings: "Women with SCA had significantly lower hemoglobin (Hb), hematocrit (Hct), and red blood cell counts, and higher mean corpuscular volume (MCV), mean corpuscular hemoglobin (MCH), and platelet count compared to controls."

The comparison between groups in terms of preterm labor and IUGR could be stated more clearly. The authors should mention that there was no significant difference in preterm labor rates (11.5% vs 7.4%, p = 0.691) but there was significant association with IUGR (OR = 4.0, p = 0.038). Furthermore, the authors should consider separately comparing small-for-gestational-age (SGA) and low-birth-weight neonates between the groups.

A typographical error is present: the correct abbreviation for neonatal intensive care unit is NICU, not NIUC. Also, Hematocrit is abbreviated as Hct, not Htc.

3. Statistical Considerations:

The authors should address the differences in maternal age and parity between the SCA and control groups. These demographic factors could confound the associations between SCA and perinatal outcomes. An adjusted analysis to account for these differences would strengthen the conclusions and help ensure that observed outcomes are primarily related to SCA rather than age or parity differences.

4. Discussion Section:

The authors provide a good overview of the pathophysiology of SCA, but they could benefit from a more detailed discussion of the specific mechanisms through which sickle cell disease impacts pregnancy outcomes. For example, how vaso-occlusive crises (VOCs) affect placental blood flow or fetal oxygenation could be explored in more depth, as this is a key factor contributing to complications like preterm birth and IUGR.

The authors mention that women with SCA are reported to have children at a more advanced age. While they report a higher average maternal age in their study, they should also compare the mean age at first pregnancy between the two groups. This would provide additional insight into the timing of reproduction in women with SCA, and how this may impact perinatal outcomes.

The study highlights IUGR and low birth weight as prevalent in SCA pregnancies. The authors should discuss the potential mechanisms contributing to these outcomes, such as poor placental perfusion, anemia, and chronic hypoxia, which could further clarify the underlying causes and risk factors for these complications.

Unfortunately, I do not have the capability to check for plagiarism.

Comments on the Quality of English Language

-

Author Response

Dear reviewer,

Thank you very much for your suggestions. We believe that our article will be much more scientific after the changes we made. We have addressed all the comments as shown in the revised manuscript in yellow highlighted.

  1. Methodology:
  • The timing of blood tests is not clearly specified. It would be helpful to know at what gestational week these tests were performed and whether this timing was consistent across both the SCA and control groups.

Answer: The blood tests were analyzed at the time of admission to the hospital for delivery, therefore, Table 2 includes gestational age at delivery and statistical information.

  • The criteria for preterm labor and intrauterine growth restriction (IUGR) are not defined.

Answer: The criteria for preterm labor and intrauterine growth restriction (IUGR) were defined.

  1. Results Section:
  • The presentation of results could be more concise. There is no need to present all the detailed statistical values from the tables in the text. A more streamlined narrative could improve readability. For example, the authors could summarize the main findings: "Women with SCA had significantly lower hemoglobin (Hb), hematocrit (Hct), and red blood cell counts, and higher mean corpuscular volume (MCV), mean corpuscular hemoglobin (MCH), and platelet count compared to controls."

Answer: The section were summarized as your suggestions.

  • The comparison between groups in terms of preterm labor and IUGR could be stated more clearly. The authors should mention that there was no significant difference in preterm labor rates (11.5% vs 7.4%, p = 0.691) but there was significant association with IUGR (OR = 4.0, p = 0.038). Furthermore, the authors should consider separately comparing small-for-gestational-age (SGA) and low-birth-weight neonates between the groups.
  • Answer: In order to determine which complication caused this difference, a pairwise comparison of rates was made.

IUGR is broadly defined as an estimated fetal weight or abdominal circumference <10th percentile for gestational age with pathological doppler findings. Those without pathological doppler findings (small-for-gestational-age) were not evaluated as complication

  • A typographical error is present: the correct abbreviation for neonatal intensive care unit is NICU, not NIUC. Also, Hematocrit is abbreviated as Hct, not Htc.

Answer: The abbreviations were corrected.

  1. Statistical Considerations:
  • The authors should address the differences in maternal age and parity between the SCA and control groups. These demographic factors could confound the associations between SCA and perinatal outcomes. An adjusted analysis to account for these differences would strengthen the conclusions and help ensure that observed outcomes are primarily related to SCA rather than age or parity differences.

Answer: An adjusted analysis was performed to account for these differences.

  1. Discussion Section:
  • The authors provide a good overview of the pathophysiology of SCA, but they could benefit from a more detailed discussion of the specific mechanisms through which sickle cell disease impacts pregnancy outcomes. For example, how vaso-occlusive crises (VOCs) affect placental blood flow or fetal oxygenation could be explored in more depth, as this is a key factor contributing to complications like preterm birth and IUGR.

Answer: The pathophysiology of VOCs and placental circulation were discussed

  • The authors mention that women with SCA are reported to have children at a more advanced age. While they report a higher average maternal age in their study, they should also compare the mean age at first pregnancy between the two groups. This would provide additional insight into the timing of reproduction in women with SCA, and how this may impact perinatal outcomes.

Answer: Unfortunately, the data we have does not include the patients' first pregnancy age data. We believe that the criticisms you have made are correct and we have added these as limitations of the study.

  • The study highlights IUGR and low birth weight as prevalent in SCA pregnancies. The authors should discuss the potential mechanisms contributing to these outcomes, such as poor placental perfusion, anemia, and chronic hypoxia, which could further clarify the underlying causes and risk factors for these complications.

Answer: The pathophysiology of poor placental circulation was discussed

Reviewer 3 Report

Comments and Suggestions for Authors

The article provides a detailed comparison of various data points between the two groups, including demographic data, laboratory findings, and fetal outcomes. The statistical analysis adds credibility to the results. The study's findings highlight the significant impact of SCA on both maternal and fetal outcomes, particularly in terms of low birth weight, neonatal intensive care unit admissions, and cesarean section rates. These findings have important implications for clinical practice. The article cites a substantial amount of relevant literature to support its findings and conclusions, enhancing the study's credibility.

This study is conducted at a single center, specifically the Mersin University Faculty of Medicine Hospital in southern Turkey. Although the sample size is relatively large, the SCA group only includes 26 pregnant women, which is relatively small and may affect the stability and representativeness of the results. The article is based on retrospective data and lacks long-term follow-up data, making it difficult to assess the long-term impact of SCA on both mothers and fetuses. The article does not thoroughly consider other potential influencing variables, such as socioeconomic status and maternal health behaviors, which could have a potential impact on the study results.

Overall, the article has strengths in its study design and data analysis but also has some limitations.

Author Response

Dear reviewer,

Thank you very much for your suggestions. We believe that our article will be much more scientific after the changes we made. We have addressed all the comments as shown in the revised manuscript in yellow highlighted.

  • The article provides a detailed comparison of various data points between the two groups, including demographic data, laboratory findings, and fetal outcomes. The statistical analysis adds credibility to the results. The study's findings highlight the significant impact of SCA on both maternal and fetal outcomes, particularly in terms of low birth weight, neonatal intensive care unit admissions, and cesarean section rates. These findings have important implications for clinical practice. The article cites a substantial amount of relevant literature to support its findings and conclusions, enhancing the study's credibility.

This study is conducted at a single center, specifically the Mersin University Faculty of Medicine Hospital in southern Turkey. Although the sample size is relatively large, the SCA group only includes 26 pregnant women, which is relatively small and may affect the stability and representativeness of the results. The article is based on retrospective data and lacks long-term follow-up data, making it difficult to assess the long-term impact of SCA on both mothers and fetuses. The article does not thoroughly consider other potential influencing variables, such as socioeconomic status and maternal health behaviors, which could have a potential impact on the study results.

Overall, the article has strengths in its study design and data analysis but also has some limitations.

Answer: Dear reviewer, thank you for your valuable suggestions. We added your suggestions to our study as limitations.

Round 2

Reviewer 1 Report

Comments and Suggestions for Authors

There is no need for further corrections. The manuscript can be accepted for publication.

Author Response

Dear reviewer, thank you for your valuable feedback on our article.